

# SICLE: a high-throughput tool for extracting evolutionary relationships from phylogenetic trees

Dan F. DeBlasio[1] and Jennifer H. Wisecaver[2,3]

[1] Department of Computer Science, University of Arizona, Tucson, AZ, United States
[2] Department of Biological Sciences, Vanderbilt University, Nashville, TN, United States
[3] Department of Ecology and Evolutionary Biology, University of Arizona, Tucson, AZ, United States

## ABSTRACT

We present the phylogeny analysis software SICLE (**Si**ster **Cl**ade **E**xtractor), an easy-to-use, high-throughput tool to describe the nearest neighbors to a node of interest in a phylogenetic tree as well as the support value for the relationship. The application is a command line utility that can be embedded into a phylogenetic analysis pipeline or can be used as a subroutine within another C++ program. As a test case, we applied this new tool to the published phylome of *Salinibacter ruber*, a species of halophilic Bacteriodetes, identifying 13 unique sister relationships to *S. ruber* across the 4,589 gene phylogenies. *S. ruber* grouped with bacteria, most often other Bacteriodetes, in the majority of phylogenies, but 91 phylogenies showed a branch-supported sister association between *S. ruber* and Archaea, an evolutionarily intriguing relationship indicative of horizontal gene transfer. This test case demonstrates how SICLE makes it possible to summarize the phylogenetic information produced by automated phylogenetic pipelines to rapidly identify and quantify the possible evolutionary relationships that merit further investigation. SICLE is available for free for noncommercial use at http://eebweb.arizona.edu/sicle/.

## INTRODUCTION

The analysis of phylogenetic trees is a critical component of evolutionary biology. Continued advances in sequencing technologies, computational power, and phylogenetic algorithms have facilitated the development of automated phylogenetic pipelines capable of quickly building hundreds of thousands of gene trees. These phylogenies can be applied to a variety of genomic problems including the functional characterization of unknown proteins (*Eisen, 1998*), orthology prediction (*Gabaldón, 2008*), and detection of gene duplication and horizontal transfer (*Huerta-Cepas et al., 2010*; *Peña et al., 2010*).

Although the ease and speed of phylogenetic pipelines continues to improve, programs for extracting the sister relationships to a clade of interest directly from gene phylogenies are rare. This is despite the utility of such information for identifying putative cases of horizontal gene transfer (HGT), hybridization, incomplete lineage sorting, and other biological processes that result in disagreement between gene and species trees. One tool that could be adapted for this purpose is The Newick Utilities, a powerful suite of Unix

Corresponding author
Dan F. DeBlasio,
deblasio@cs.arizona.edu

shell programs for processing phylogenetic trees (*Junier & Zdobnov, 2010*). However, the suite works best when processing species phylogenies; trees must be rooted, although rerooting is possible. This makes it difficult to automate the analysis of gene phylogenies in which the biological root is unknown (e.g., many bacterial trees). Another strategy for the high-throughput parsing of phylogenies is to search for a predefined association of interest; for example, interdomain HGT between co-occurring extremophilic bacteria and archaea (*Nesbø et al., 2009*) or HGT of cyanobacterial origin into the genomes of green algae (*Moustafa & Bhattacharya, 2008*). Several programs have implemented similar search processes including PhyloSort (*Moustafa & Bhattacharya, 2008*), Pyphy (*Sicheritz-Pontén & Andersson, 2001*), PhyloGenie (*Frickey & Lupas, 2004*) and most recently PhySortR (*Stephens et al., 2016*). However, in order to identify and summarize multiple evolutionary signals in a set of gene trees (e.g., all putative cases of HGT, even from unanticipated donors), one must manually iterate through such programs to identify all possible sister associations to the target clade of interest. Similar functionality can be found in new tools such as the ETEToolkit (*Huerta-Cepas, Serra & Bork, 2016*) but requires some expertise in python programming to integrate into a given phylogenetic pipeline, and the user must build their own wrapper to automate the detection of sister associations to a clade of interest.

We present the phylogeny analysis software SICLE (**Si**ster **Cl**ade **E**xtractor), a tool to identify the nearest neighbors to a node of interest in a phylogenetic tree as well as the support value for the relationship. With SICLE it is possible to summarize the phylogenetic information produced by automated phylogenetic pipelines for the rapid identification and quantification of possible evolutionary relationships that merit further investigation. The program is a convenient command line utility and is easy to adapt and implement in existing phylogenetic pipelines. In addition, the classes provided by SICLE can be used as an API to provide a building block for new applications, which is similar to the functionality given to the user in the ETEToolkit (*Huerta-Cepas, Serra & Bork, 2016*) but in C++ for those that prefer this programming language. In the next section, we outline our approach and briefly describe the implementation methods. The source code and example for SICLE are available for download at http://eebweb.arizona.edu/sicle/ free for noncommercial use under a Creative Commons Attribution-NonCommercial-ShareAlike (CC BY-NC-SA) License. We conclude by showing the benefit of SICLE by identifying horizontal gene transfer in *Salinibacter ruber* previously studied by *Mongodin et al. (2005)* and *Peña et al. (2010)*, not only replicating their result but identifying additional HGT candidates present in the published phylogenies.

## METHODS

The program is a simple command line utility written as a set of C++ classes. The program accepts single tree files in newick format and outputs the label of the sister association(s) to a specified clade of interest as well as any branch support for the association in a parseable, tab-separated format. The program requires that the leaf names begin with a group identifier followed by a hyphen. This identifier can correspond to a rank in the

taxonomic hierarchy (e.g., bacterial phyla) as well as other, custom classification schemes to fit the needs of individual projects. Because each tree is processed independently, the running time should be linear as the number of trees being processed increases (without considering the filesystem constraints). Trees can be processed in parallel as there are no interdependencies between the analysis of different trees in the same experiment. The process that SICLE follows has 3 major steps:

(1) Identify the target subtree. The node at the lowest common ancestor of all target leaves represents a subtree, which could consist of a single leaf. The *target leaf* or *leaves* are identified in one of two ways given the search string $S$: (a) by using an exact prefix match (i.e., we assume that the first $|S|$ characters of the leaf match those in $S$) or (b) by considering $S$ to be a regular expression (using the –E command line argument) and testing if the leaf label satisfies the given expression. The target subtree is located as follows: given a search string $S$, find the node $v$ in the tree (if one exists) for which every leaf in the subtree is a target leaf. If the target leaves are divided the program attempts to find a new root such that $v$ exists. If no re-rooting will make the target leaves monophyletic the program halts and alerts the user that a target clade cannot be found. Therefore, SICLE may be unable to detect horizontally acquired gene copies (i.e., xenologs) if the ancestral gene copy is retained and also present in the phylogeny, because the target may not be monophyletic. In these cases, the user can reanalyze the subset of trees in which the target is non-monophyletic manually and/or by specifying a different target search string. The search string $S$ as a prefix is flexible and can correspond to a specific group identifier (e.g., Bacteroidetes), a subgroup (e.g., Bacteroidetes-Salinibacter), or even an individual leaf node (e.g., Bacteroidetes-Salinibacter_ruber_Phy001XKJS). This search criteria becomes even more flexible when the labeling is done using a regular expression.

(2) Identify the subtrees of the possible sisters to the target. SICLE assumes that the root is arbitrary and that an outgroup is not necessarily known or available. Therefore, SICLE returns up to two possible sisters to the target (S1 and S2, Fig. 1). These associations are contingent on two alternative tree rootings and are mutually exclusive. SICLE leaves it up to the user to determine which is the more biologically consistent sister association, if possible. One way to do this is to rank the possible sister associations based on their phylogenetic distance from the target based on the species tree (*Wisecaver & Hackett, 2014*). Determining the possible sisters falls into two cases: (2a) When the target subtree is a child of the root, the two sisters are the two children of the other child of the root (Fig. 1A). (2b) When the target subtree is not a direct descendant of the root, the other child of the target's parent is one sister and the rest of the phylogeny is considered the other sister, as if the tree is re-rooted at the parent of the target subtree (Fig. 1B).

(3) Determine if a sister subtree corresponds to a distinct taxonomic unit. The final step follows the same search procedure as step one. SICLE determines if all leaves of a sister subtree have the same group identifier, and if so returns the group identifier and the branch support for the parent node uniting the target and sister subtrees. A hierarchical grouping of identifiers can be specified to expand the results and customize them for any project. For example, if the group identifiers were to correspond to plant and fungal divisions and animal phyla, the configuration file could classify these identifiers into the

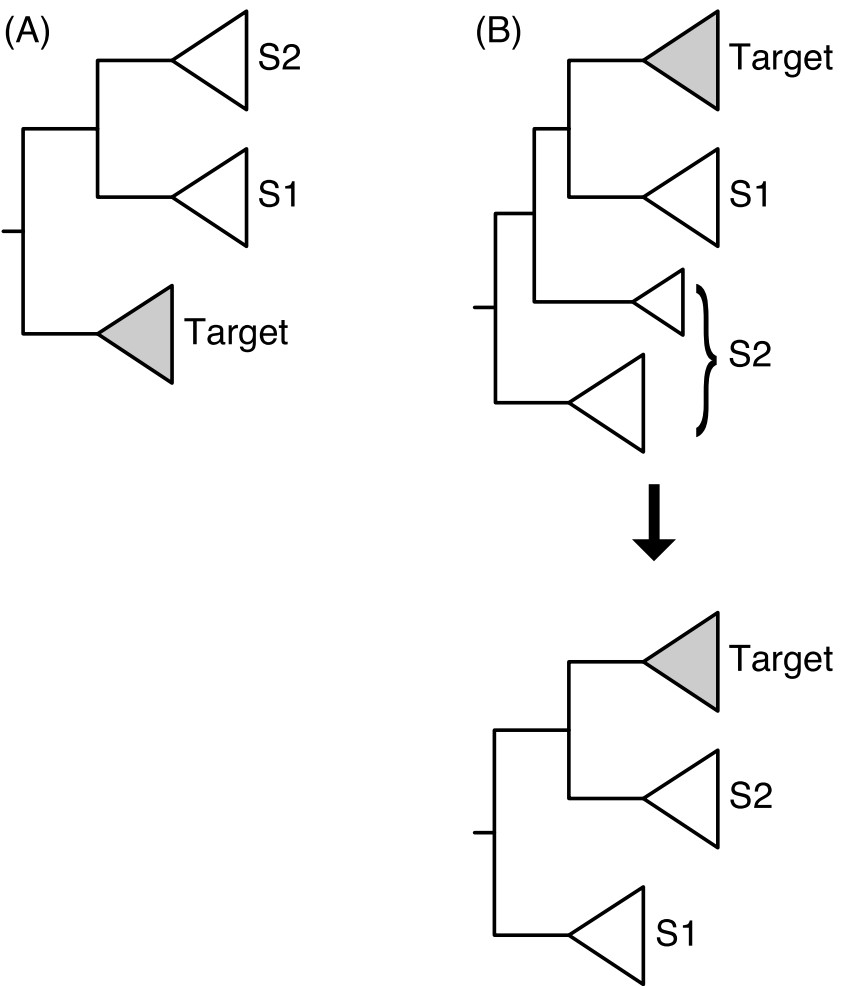

**Figure 1   Two configurations for the identification of the sister subtrees given the location of the target subtree.** In (A) the target subtree is a direct descendant of the root of the tree, and in (B) it is not. Note that in (B) the tree can be rerooted visually even though this is not performed in practice.

kingdoms Plantae, Fungi, and Animalia. Animalia and Fungi could be further categorized as Opisthokonta, and all three are Eukaryota. An example configuration file is available on the SICLE website. The hierarchy must be properly nested; however, it is possible to assess the results from alternative, conflicting hierarchies by rerunning SICLE specifying different configuration files. When a group configuration file is given, SICLE identifies the smallest hierarchical class that can summarize the whole sister subtree. If both sisters belong to the same hierarchical group, they are combined to return only a single result. If a sister clade does not fall into a single class in the hierarchy, no result is produced.

## CASE STUDY

The utility of SICLE was demonstrated using gene trees from the halophilic Bacteroidetes *Salinibacter ruber*. Several cases of inter-domain HGT from halophilic archaea were previously identified in two published genomes from strains M8 and M13 (*Peña et al., 2010*;

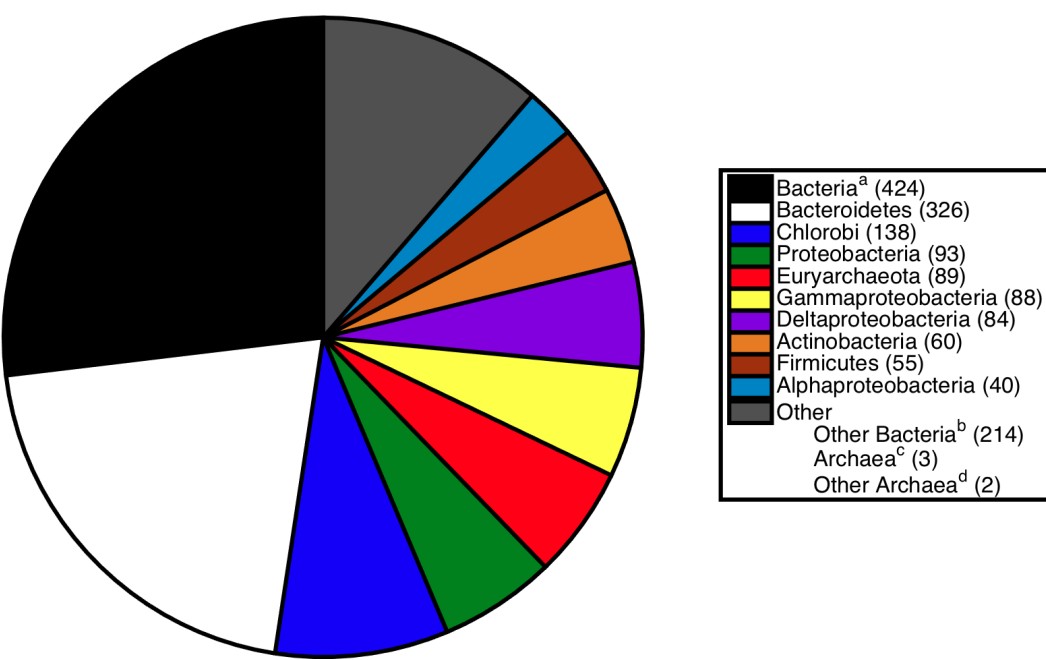

**Figure 2** **Breakdown of sister relationships to the subtree for *S. ruber* in 2,315 gene trees generated for strain M8.** [a]Bacteria, the sister subtree contained more than one bacterial phyla. [b]Other Bacteria, the sister consisted of a single bacterial phyla not already listed above. [c]Archaea, the sister subtree contained more than one archaeal phyla. [d]Other Archaea, the sister consisted of a single archaeal phyla other than Euryarchaeota.

*Mongodin et al., 2005*). The trees were downloaded from `PhylomeDB`, a public database containing complete collections of gene phylogenies for organisms (*Huerta-Cepas et al., 2011*). A `BioPerl` script was used to prepend group identifiers to leaf names. These prefixes corresponded to prokaryotic phyla, except in the case of the proteobacterial leaves, which were prefixed with class identifiers (e.g., Gammaproteobacteria). The `BioPerl` script is available on the `SICLE` website.

A total of 2,315 and 2,274 gene phylogenies were analyzed from *S. ruber* M8 and M13 respectively. Trees were first parsed using the search prefix 'Bacteroidetes-Salinibacter _ruber' to identify 1,463 (M8) and 1,457 (M13) trees (from 1,499 orthologous clusters) in which the two strains were monophyletic. Trees in which *S. ruber* was not monophyletic were further parsed using search prefixes corresponding to M8 or M13 alone, and sister(s) to individual strains were identified in 91 (M8) and 72 (M13) additional phylogenies. In tests `SICLE` is able to analyze just over 3,000 phylogenies per minute on a 2.53 GHz Core 2 Duo laptop computer with 8GB of available RAM (though `SICLE` used less than 1 GB).

The breakdown of sister associations to *S. ruber* present in strain M8 trees is shown in Fig. 2. The most common sister was Bacteria, a higher level classification indicating the sister clade consisted of two or more bacterial phyla. The next most abundant sisters were Bacteriodetes (326 trees) and Chlorobi (138 trees). These associations were anticipated and suggest vertical gene inheritance, because *S. ruber* is a member of the Bacteriodetes/Chlorobi superphylum. Other common bacterial sisters included members

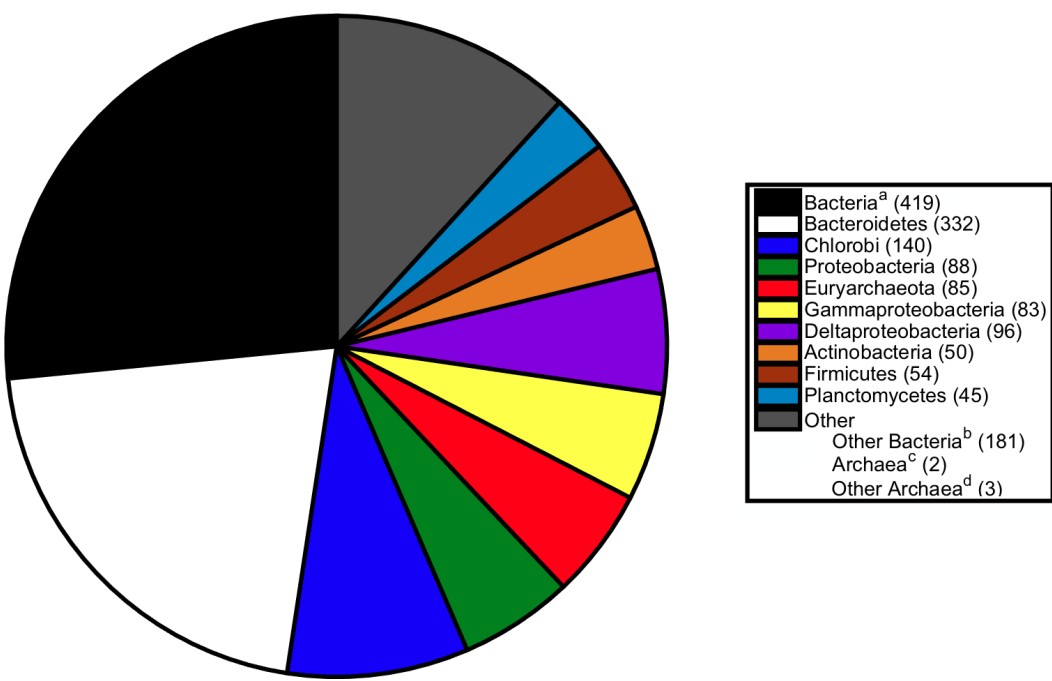

**Figure 3** Breakdown of sister relationships to the subtree for *S. ruber* in 2,274 gene trees generated for strain M13. [a]Bacteria, the sister subtree contained more than one bacterial phyla. [b]Other Bacteria, the sister consisted of a single bacterial phyla not already listed above. [c]Archaea, the sister subtree contained more than one archaeal phyla. [d]Other Archaea, the sister consisted of a single archaeal phyla other than Euryarchaeota.

of the Proteobacteria, Actinobacteria, and Firmicutes (Fig. 2). The previously published association between *S. ruber* and the archaeal group Euryarchaeota was recovered in 89 gene phylogenies. Of those, 74 gene phylogenies showed *S. ruber* as exclusively sister to Archaea (i.e., the trees could not be rerooted in such a way that *S. ruber* had a second possible sister association to bacteria or eukaryotes), and of those, 33 gene phylogenies showed high branch support for the association between *S. ruber* and Euryarchaeota (PhyML local support ≥0.90). The proportion of sister associations present in strain M13 were virtually identical to those found in M8 (Fig. 3).

## DISCUSSION

A common and successful application of automated phylogenetic pipelines is for the estimation of HGT based on phylogenetic incongruence between gene phylogenies and an accepted species tree (*Sicheritz-Pontén & Andersson, 2001*). However, prior to tree building, many studies first select candidate genes suspected of being horizontally acquired based on local sequence similarity to possible donor lineages (e.g., *Peña et al., 2010*; *Gladyshev, Meselson & Arkhipova, 2008*; *Maruyama et al., 2009*; *Nowack et al., 2011*), which we refer to as a BLAST-first approach. In these analyses, phylogenetics is used *to confirm* cases of HGT rather than actually identify putative transfers. The need to restrict the number of trees in an analysis has little to do with the computational requirements of the phylogenetic methods, but is rather to minimize the number of phylogenies that then require manual

inspection, a significant time investment. This BLAST-first approach is susceptible to two types of error: (1) genes selected for further analysis based on local similarity may not support the prediction of HGT once a phylogenetic model of evolution is applied to the global alignment (*Koski, Morton & Golding, 2001*), and (2) genes whose phylogenies would be indicative of HGT may not pass the BLAST similarity thresholds (e.g., E-value, percent coverage) and are thus not analyzed (*Wisecaver & Hackett, 2014*). Without an efficient method for identifying associations suggestive of HGT directly from phylogenies, the alternative to the BLAST-first approach is manual inspection of each gene tree. For example, in a recent study of HGT from fungi in the plant-pathogenic oomycetes, the authors opted to manually inspect all 11,434 phylogenies for cases of gene transfer rather than limit their analysis to oomycete genes with a high BLAST hit to fungi (*Richards et al., 2011*).

*Peña et al. (2010)* identified genes putatively involved in interdomain HGT between *S. ruber* and Archaea. In their analysis, genes were first screened for a best BLAST hit to archaeal genes with E-values below E-20 and a minimum query sequence overlap of 85%. Using this BLAST-first approach, the authors identified 40 candidate genes in *S. ruber* strain M8 putatively acquired from Archaea. Further validation of possible gene transfer was then performed using an analysis of oligonucleotide frequencies. Of the 40 candidate genes, 23 had gene phylogenies available on PhylomeDB, and only 8 supported the BLAST-determined association between *S. ruber* and Archaea with PhyML local support $\geq 0.9$. With SICLE, we identified an additional 25 gene phylogenies showing an exclusive association between *S. ruber* and Archaea, information that was parsed directly from the gene phylogenies rather than being first filtered based on local similarity. These 25 gene phylogenies may provide additional examples of HGT between these two groups and warrant further investigation. For example, the gene phylogeny Phy001XM22, a cation transport protein family, shows *S. ruber* grouping sister to five species of halophilic Euryarchaeota with strong support (Fig. 4).

It is not our intent to suggest that all the trees identified by SICLE that group *S. ruber* together with Archaea necessarily demonstrate true cases of HGT. On the contrary, there are many other possible sources of atypical phylogenetic placement, including contamination (*Wisecaver et al., 2016*), taxon sampling (*Rokas et al., 2003*), long branch attraction (*Brinkmann et al., 2005*), incomplete lineage sorting (*Ebersberger et al., 2007*), and differential gene loss (*Qiu et al., 2012*). Rather than the endpoint of a phylogenetic analysis, the purpose of SICLE is to quickly and efficiently summarize the patterns present in large collections of gene phylogenies. Just as putative cases of HGT can be identified via BLAST (*Gladyshev, Meselson & Arkhipova, 2008*), stochastic mapping (*Cohen & Pupko, 2010*), and compositional attributes (*Lawrence & Ochman, 1998*), SICLE identifies putative cases of HGT based on tree topology. In addition to identifying potential HGT, SICLE can be easily adapted to other phylogenetic problems requiring the quantification of evolutionary signals present in gene trees including endosymbiosis, hybridization, and incomplete lineage sorting.

We have presented a new tool for high-throughput phylogenetic analysis, SICLE (**Si**ster **Cl**ade **E**xtractor). With this tool users are able to quickly identify trees with interesting relationships that can be prioritized for further analysis. The ability to identify subtrees

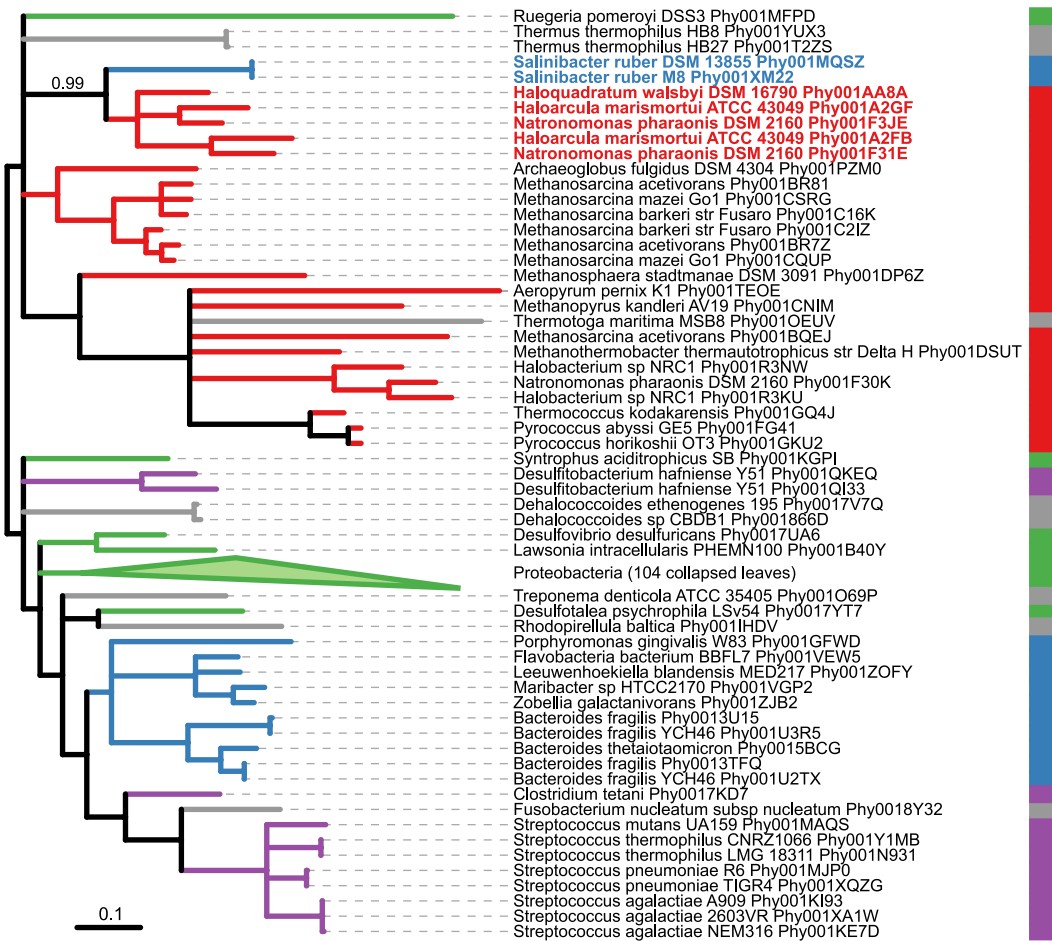

**Figure 4** Maximum likelihood phylogeny `Phy001XM22` from `PhylomeDB` showing possible HGT from Euryarchaeota to *S. ruber*. The phylogeny has been midpoint rooted and visualized using `iTOL` version 3.0 (*Letunic & Bork, 2016*). Branches with PhyML local support less than 0.90 were collapsed using `TreeCollapserCL` version 4.0 (*Hodcroft, 2015*). Branches are colored according to taxonomy: Bacteroidetes, blue; Archaea, red; Proteobacteria, green; Firmicutes, purple; other Bacteria, grey. The two *S. ruber* sequences are bolded and colored blue, and the five sister Euryarchaeota sequences are bolded and colored red. The local support for the association is indicated above the branch.

using a regular expression and having a tiered sister labeling system allows for very complicated searches to be preformed. While `SICLE` is a powerful analysis tool it is meant as a filtering mechanism for later analysis, i.e., the first step in making an observation about a large number of phylogenies. In the case study, we showed the usefulness of `SICLE` for identifying horizontal gene transfer. We showed that by using `SICLE` instead of a pre-tree `BLAST` search, additional putative HGTs are identified while still recovering all of the previously found results.

## ACKNOWLEDGEMENTS

We are grateful to Andy Gloss and John Kececioglu for reviewing the manuscript and providing helpful feedback.

### Funding

This work was funded by the NSF IGERT in Genomics at the University of Arizona DGE-0654435. The funders had no role in study design, data collection and analysis, decision to publish, or preparation of the manuscript.

### Grant Disclosures

The following grant information was disclosed by the authors:
NSF IGERT in Genomics at the University of Arizona: DGE-0654435.

### Competing Interests

The authors declare there are no competing interests.

### Author Contributions

- Dan F. DeBlasio conceived and designed the experiments, performed the experiments, contributed reagents/materials/analysis tools, wrote the paper, prepared figures and/or tables, reviewed drafts of the paper.
- Jennifer H. Wisecaver conceived and designed the experiments, performed the experiments, analyzed the data, contributed reagents/materials/analysis tools, wrote the paper, prepared figures and/or tables, reviewed drafts of the paper.

### Data Availability

All source code and data is available at http://eebweb.arizona.edu/sicle/.

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
