# Peer review of "SICLE: a high-throughput tool for extracting evolutionary relationships from phylogenetic trees"

_PeerJ, doi:10.7717/peerj.2359_

## Round 0.1 · original submission · Major Revisions

As the reviewers note there are several major problems with this paper as it stands. Overuse of hyperbole: e.g. "...easily parseable, tab-separated format" when tab-separated format alone would suffice). The quality of the software is questionable: e.g. embedding parameters into leaf names, lack of a -h option, a general lack of attention to the documentation that is needed for any software to have an impact beyond a local utility, and of course the licensing issue is glossed over. Finally the contradictory numbers need to be resolved and clarified, and the information gaps must be filled in to complete this paper.

Reviewer 1 ·

Basic reporting

Introduction. ETE (etetoolkit.org) is another important set of tools for large-scale analysis of phylogenetic trees.

It would be nice to find the software availability information at the end of the abstract.

Experimental design

see below

Validity of the findings

see below

Additional comments

The manuscript describes a command-line utility for the large-scale extraction of evolutionary gene relationships from phylogenetic trees, suitable for integration into existing analysis pipelines.

1. Method (page2/60) & Figure 1. It remains unclear whether trees are treated as rooted or not. If the root is considered ‘arbitrary’ (page2-60), how is the sister clade determined? A clear description of this essential analysis step is missing in the Methods section.

Similar: The figure legend states that the tree is “rerooted visually even though this is not performed in practice”; however, the tree in Figure 1B is definitely rerooted.

The last sentence of the Figure 1 legend is probably obsolete.

2. S. ruber example: large-scale tree inference can be error-prone, e.g. due to erroneous data; the study lacks evidence that the additional 54 HGT predictions are actually not false positives.

3. It is stated (page 2) that trees with non-monophyletic target leaves are not processed by SICLE. How are xenologs identified if a species retained the ortholog?

·

Basic reporting

The manuscript presents a software tool to identify the sister clade of a particular clade of interest. The tool is written in C++ and the code is available, though the license is unclear.

The tool adopts the UNIX philosophy of doing one single thing. However, it is not particularly feature-rich. The interface is spartiate and the documentation quasi-inexistent. This latter point should be addressed in a revised version of the submission.

The manuscript contains many self-aggrandising claims about being "easy-to-use", "adaptable", "very powerful", which are vague and largely unsubstantiated. Given PeerJ's overriding editorial criterion of correctness, not significance, these claims are not so relevant anyway so I recommend dropping them altogether.

Specific points:

- Abstract: the tool is says "easy-to-use, adaptable". Two lines later "convenient line utility and is easy to adapt". Remove the redundancy, and explain in which way the tool is adaptable.

- Caption "Biological examples of target and sister clades are provided in parentheses." Where are these parentheses?

- "These associations were anticipated are suggest vertical gene inheritance". This phrase has a problem. Rewrite.

- "A hierarchical grouping of identifiers can be specified to expand the results 82 and customize them for any project.". This function is not documented.

- "The proportion of 117 sister associations present in strain M13 were virtually identical to those found in M8 (data not shown)". Why not show these data? PeerJ is online-only and the article is short so there is no reason not to include this result.

- "With SICLE, we identified over twice the number (94 trees) of potential 143 gene transfers from Archaea in strain M8.". Is it now 89 or 94?

- "Our results suggest that 145 over half (54 of 94) of the gene trees that show S. ruber grouping with Archaea were missed using a 146 BLAST-first approach. No evidence is provided that these "missing" gene trees are not, in fact, incorrect.". Next paragraph "It is not our intent to suggest that all the trees identified by SICLE that group S. ruber together 148 with Archaea necessarily demonstrate true cases of HGT". Which one is it then? It would be good to provide representative examples of these contentious cases.

- "website The hierarchy". Full stop missing.

- "The ability to identify subtrees using a regular expression and having a tiered sister labeling system allows for very complicated searches to be preformed". The regular expression support is entirely undocumented.

Experimental design

Not applicable.

Validity of the findings

- The tool mention use of bootstrap support value, but as far as I can tell, there is no attempt to account for bootstrap support in fig 2. What sort of bootstrap support is associated with the 89 putative HGT instances reported in this analysis?

- "very powerful analysis tool". This is an overstatement.

- "In the case study, we showed the usefulness of SICLE for identifying horizontal gene transfer". This needs to be toned down.

Additional comments

Comments on source code:
* * *
- The code compiles fine but some checks are needed. For instance, running the code without input results in a segmentation fault.... the standard -h option for help is unsupported.

- There is no mention of the license on the homepage or as part of the code tarball.

- The readme files has riddled with typos (e.g. tis, polyphioletic, manulally, alternatly, wihch)

- Please include an example file.

---

## Round 0.2 · Minor Revisions

Please ensure the locations and downloads mentioned by reviewer 1 are working correctly.

Reviewer 1 ·

Basic reporting

Re-Review: All my previous concerns have been addressed in the updated manuscript.

Experimental design

-

Validity of the findings

-

Additional comments

Note:
The page "http://eebweb.arizona.edu/sicle" is not available; the URL only works only when ending with a backslash: http://eebweb.arizona.edu/sicle/

The download at http://eebweb.arizona.edu/sicle/SiClE_v1.2_x86-64.tgz is not available; the correct address should be
http://eebweb.arizona.edu/sicle/SiClE_v1.2_x86_64.tgz (x86underscore64)

There seem to exist problems with precompiled download files (corrupted, incomplete?)

·

Basic reporting

My points were all satisfactorily addressed and I have no further reservation.

Experimental design

Not applicable.

Validity of the findings

Not applicable.

Additional comments

My points were all satisfactorily addressed and I have no further reservation.

---

## Round 0.3 · accepted · Accept

I appreciate your patience throughout this process.